# Nitroxyl Delivered by Angeli’s Salt Causes Short-Lasting Activation Followed by Long-Lasting Deactivation of Meningeal Afferents in Models of Headache Generation

**DOI:** 10.3390/ijms23042330

**Published:** 2022-02-19

**Authors:** Stephanie K. Stöckl, Roberto de Col, Milos R. Filipovic, Karl Messlinger

**Affiliations:** 1Institute of Physiology and Pathophysiology, Friedrich-Alexander-University Erlangen-Nürnberg, 91054 Erlangen, Germany; moooh-ment@web.de (S.K.S.); roberto.col@fau.de (R.d.C.); 2Leibniz Institute for Analytical Sciences, ISAS e.V., 44227 Dortmund, Germany; milos.filipovic@isas.de

**Keywords:** TRPA1 receptors, nitroxyl, acrolein, primary meningeal afferents, spinal trigeminal neurons, trigeminal nociception, headache, mouse, rat

## Abstract

The role of TRPA1 receptor channels in meningeal nociception underlying the generation of headaches is still unclear. Activating as well as inhibitory effects of TRPA1 agonists have been reported in animal models of headache. The aim of the present study was to clarify the effect of the TRPA1 agonist nitroxyl (HNO) delivered by Angeli’s salt in two rodent models of meningeal nociception. Single fibre recordings were performed using half-skull preparations of mice (C57BL/6) in vitro. Angeli’s salt solution (AS, 300 µM) caused short-lasting vigorous increases in neuronal activity of primary meningeal afferents, followed by deactivation and desensitisation. These effects were similar in TRPA1 knockout and even more pronounced in TRPA1/TRPV1 double-knockout mice in comparison to wild-type mice. The activity of spinal trigeminal neurons with afferent input from the dura mater was recorded in vivo in anesthetised rats. AS (300 µM) or the TRPA1 agonist acrolein (100 and 300 µM) was applied to the exposed dura mater. AS caused no significant changes in spontaneous activity, while the mechanically evoked activity was reduced after acrolein application. These results do not confirm the assumption that activation of trigeminal TRPA1 receptor channels triggers the generation of headaches or contributes to its aggravation. Instead, there is evidence that TRPA1 activation may have an inhibitory function in the nociceptive trigeminal system.

## 1. Introduction

Transient receptor potential (TRP) channels comprise a big family of seven subfamilies with several subtypes, all of them unspecific cation channels with six transmembrane domains [1]. Primary nociceptive afferents express TRP channels of the vanilloid subfamily (TRPV), particularly the subtype TRPV1, the first cloned TRP channel, and the ankyrin family with its only known subtype TRPA1, characterised by a big intracellular domain with 17 ankyrin repeats [2]. These two TRP types are still in the focus of scientific interest, not least through the recent Nobel prize for their discovery [3]. They are frequently co-expressed, for example, in meningeal afferents [4] and are partly responsible for the release of neuropeptides such as calcitonin gene-related peptide (CGRP) from nociceptive terminals [5], a mechanism that has considerable significance for meningeal nociception and headache generation [6].

TRPA1 agonists such as acrolein activate the receptor channel through modification of reactive cysteine residues of the intracellular N-terminal [7,8]. Endogenous TRPA1 ligands are metabolites such as H_2_O_2_ and 15d-PGJ_2_ that are generated under oxidative stress; experimentally, they can cause pain responses that are lacking in mice with functional knockout of TRPA1 receptors [9]. An environmental TRPA1 agonist named umbellulone, the scent of the Californian laurel, has been reported to causes an increase in meningeal blood flow when inhaled by rats and can cause headaches in sensitive persons [10]. Besides acrolein used in the present study, another potent TRPA1 agonist may be nitroxyl (HNO), an unstable redox sibling of nitric oxide (NO). Nitroxyl has been shown to release CGRP from trigeminal and other afferents increasing meningeal blood flow, which could be blocked by the TRPA1 receptor antagonist HC030031 [11]. Nitroxyl reacts preferably with thiol groups thus activating TRPA1 at its intracellular cysteine residues [12]. The mostly used donor of nitroxyl is Angeli’s salt (Na_2_N_2_O_3_), which is stable in alkaline solution under nitrogen atmosphere [13]. Upon protonation in neutral solution, Angeli’s salt decomposes fast to liberate nitroxyl (HNO).

Apart from the TRPA1-activating effect causing neuropeptide release upon calcium inflow, it is not clear if nitroxyl activates primary (meningeal) afferents and hence second-order neurons, which depends on the generation of propagated activity in primary afferents. We set out to clarify this matter using two models of meningeal nociception—first, recordings from meningeal afferents in the hemisected mouse skull in vitro, and second, recordings from neurons in the rat spinal trigeminal nucleus in vivo. Mice with genetically deleted TRPA1 receptors as well as TRPA1/TRPV1 double-knockout animals were used to control for the specificity of nitroxyl-TRPA1 effects.

## 2. Results

### 2.1. Primary Afferent Recordings In Vitro

Recordings of afferent activity were made in 27 skull halves of 19 mice (C57BL/6: 9/7; TRPA1^−/−^: 9/7; TRPA1/V1^−/−^: 9/5) either from the tentorius nerve (*n* = 31) or from the spinosus nerve (*n* = 7). The tentorius nerve arising from the ophthalmic division of the trigeminal ganglion innervates the tentorium cerebelli and extends near the sagittal sinus to the frontal region of the middle cranial fossa, innervating large parts of the upper parietal dura mater (Figure 1A). The spinosus nerve arising from the mandibular division consists of 2–3 nerve bundles innervating the basal region of the middle cranial fossa; they are more difficult to identify than the tentorius nerve. The mechanical receptive fields were distributed over the whole dural lining but most frequently located in the dorsal region of the middle cranial fossa. At these points, action potentials could be elicited with electrical thresholds ranging from 3 to 38 µA. The conduction velocities of afferent fibres calculated from the latency and the conduction distance ranged from 0.8 to 13.3 m/s, and thus the afferents were in the C-fibre (*n* = 12) or in the Aδ-fibre range (≥2.5 m/s, *n* = 25). In all three genotype groups, Aδ-fibres formed the majority of afferents. The resting activity of afferents (without electrically stimulated activity) was very low during the control periods in all three groups, ranging between 0 and 8 impulses/min, and in most cases it was 0. Comparing the activity within the 5 min intervals before stimulation with AS, we found some variation but no significant difference between the intervals or between the genotype groups (three-way ANOVA with repeated measures, extended by Tukey’s post hoc test, *p* > 0.05). 

In most experiments, one afferent fibre was recorded, but in three experiments two, in two experiments three, and in one experiment four fibres with discriminable action potentials were recorded in parallel and separately processed off-line. Figure 1B shows a short segment of an example recording with two fibres and their responses to electrical and chemical stimulation. The application of Angeli’s salt (AS) was followed by a cluster of discharges within a few seconds. This was visibly similar in all experiments. Stopping of the circulation without application of substances or application of vehicle (NaOH) did not cause significant increases in action potential firing.

#### 2.1.1. Responses to AS in C57BL/6 Wild-Type Mice

In 10 of 13 afferents, the activity increased within the first min after AS application, on average by more than six times of the activity within the control period at the beginning of the experiment, while in two afferents, the maximal activity was reached in the second minute. After these action potential clusters, the activity decreased rapidly, staying below the control activity; at the latest from the fourth minute after AS, 12 of the 13 afferents were no longer able to be electrically activated with constant stimulation strength. One fibre showed no increase in activity but lost its responsiveness immediately after AS application. Comparing the 5 min intervals with repeated measures ANOVA, the changes during the experiments are highly significant (*F*_9,108_ = 21.0; *p* < 0.0001). The mean activity within the 5 min stimulation period with AS increased by 2.5 times compared to the control activity during the first 5 min (Figure 2). The increase in activity was different to the first control period (Tukey, *p* = 0.013) but failed to reach statistically significance compared to the other 5 min intervals prior to stimulation (*p* = 0.11–0.38), whereas the decrease in activity after AS stimulation is highly significant (*p* < 0.0005).

#### 2.1.2. Responses to AS in TRPA1^−/−^ Mice

In 11 of 12 afferents, the activity increased within the first min after AS application, on average nearly 12 times of the initial control activity. In 10 fibres, the highest activity was reached within the first minute after AS application; in one fibre within the second min; and in one fibre, it was delayed to the fifth min. Following the maximum, the activity rapidly decreased to the baseline or below. Only three fibres remained responsive to electrical stimulation after the stimulation interval. Comparing the 5 min intervals with repeated measures ANOVA, we found that the changes were highly significant (*F*_9,99_ = 13.52; *p* < 0.0001). The mean activity within the 5 min stimulation period increased by 4.5 times compared to the initial control. The increase in activity was different to all 5 min periods prior to and after AS application with high significance (Tukey’s test, *p* < 0.0005).

#### 2.1.3. Responses to AS in TRPA1/V1^−/−^ Mice

In 8 of 13 afferents, the activity increased within the first min after AS application, on average nearly 15 times of the initial control activity. In 11 afferents, the highest activity was reached within the first minute after AS application, in one fibre in the second and in one within the third min, before the activity fell to baseline or below. Only five fibres remained responsive to electrical stimulation after AS stimulation. Comparing the 5 min intervals with repeated measures ANOVA, we found that the changes were highly significant (*F*_9, 108_ = 9.21; *p* < 0.0005). The mean activity within the 5 min stimulation period increased by about 4.4 times compared to the initial control. The increase in activity was different to all 5 min periods prior to and after AS application with high significance (Tukey’s test, *p* < 0.0005).

### 2.2. Second-Order Neuron Recordings In Vivo

Recordings were made from the spinal trigeminal nucleus in 22 rats. Only one neuron per experiment and animal was recorded. Recording sites were located 1.3–2.8 mm (median 2.2 mm) caudal from the obex, 1.1–2.3 mm (median 1.35 mm) from the midline, and in a depth of 342–1311 µm (median 751 µm) from the medullary surface (Figure 3). All units had a receptive field in the dura mater at various positions, wherein they could be activated with von Frey stimuli of 6.9–14.7 mN and with electrical stimuli of 0.16–3.9 mA. The latency at threshold was 8–25 ms, indicating afferent input by C- and Aδ-fibres. All but one unit had facial receptive fields in the ipsilateral ophthalmic (V1) or the maxillary (V2) region; all but five units also had input from the ipsilateral temporal muscle, and eight units responded to gentle mechanical stimulation of the cornea. There was no significant difference of these general properties between the group of neurons challenged with Angeli’s salt or with acrolein, respectively.

#### 2.2.1. Responses of Spinal Trigeminal Neurons to Angeli’s Salt

In 11 experiments (11 neurons), Angeli’s salt (300 µM) was tested. The spontaneous activity of neurons ranged from 0 (1 neuron) to 1065 impulses/min (mean 244.6 ± 110.9 imp/s) during the control period of 5 min, which was followed by topical application of SIF, NaOH (5 mM), SIF, Angeli’s salt (300 µM), and finally SIF as post-control. Mean activity (impulses/min) within these 5 min periods was assessed, and the interposed washing sections of 1–2 min were discarded. The ongoing activity was not significantly different from baseline (100%) in any of the application periods (repeated measures ANOVA, *F*_5,50_ = 1.56; *p* = 0.19; Figure 4A). The mechanically evoked activity during the 1 s stimulation period (ongoing activity subtracted) ranged from 2.4 to 43.5 imp/s (mean 16.5 ± 4.2 imp/min) and did not significantly change for the duration of the whole experiment (*F*_5,50_ = 1.00; *p* = 0.43; Figure 4B). Analysing 1 min intervals instead of 5 min intervals, we found that there was again no significant change in ongoing or evoked activity.

#### 2.2.2. Responses of Spinal Trigeminal Neurons to Acrolein

In 11 other experiments (11 neurons), the TRPA1 agonist acrolein was tested. The spontaneous activity of neurons ranged from 1.4 to 1455 impulses/min (mean 394.8 ± 135.8 imp/min) during the control period of 5 min, which was followed by topical application of SIF, acrolein 100 µM, short washing, acrolein 300 µM, and finally two SIF periods as post-control. Mean activity (impulses/min) within these 5 min periods was assessed, and interposed washing sections of 1–2 min were discarded. The ongoing activity did not significantly change in any of the application periods, although there was a tendency towards an increase (repeated measures ANOVA, *F*_7,56_ = 1.46; *p* = 0.20; Figure 5A). Analysing 1-min sections, we also found no significant change in ongoing activity. The mechanically evoked activity during the 1 s stimulation period (ongoing activity subtracted) ranged from 7.6 to 38.1 imp/s (mean 20.0 ± 4.0 imp/s). ANOVA indicated significant variation during the experiment (*F*_7,49_ = 2.55; *p* = 0.025); according to Tukey’s post hoc test, the activity within the acrolein 100 µM application period (*p* < 0.05) and the 300 µM application period (*p* < 0.01), as well as the first post-control period (*p* < 0.05), were lower than within the control period at the beginning of the experiment (Figure 5B).

## 3. Discussion

Nitroxyl (NO^−^ or HNO) has previously been shown to release the vasodilator neuropeptide CGRP from primary meningeal afferents and to thus increase meningeal blood flow in rodents [11,14]. In the present study, we used Angeli’s salt (AS) as a potent nitroxyl donor [13] to probe if AS is also capable of activating murine primary meningeal afferents in terms of eliciting propagated action potentials, and secondly, if this may activate central neurons in the rat spinal trigeminal nucleus with meningeal afferent input. Following a short-lasting vigorous activation of meningeal afferents, AS caused a long-lasting desensitisation and depression of primary afferents. Assuming that these effects are dependent on TRPA1 activation, we used TRPA1-deficient mice, but unexpectedly, the AS effects in this genotype were not only similar to the wild-type mice but rather pronounced. Speculating that AS may exert cross-activation and deactivation of TRPV1 receptors, we used TRPA1/TRPV1 double-knockout mice but received very similar results. To figure out if the primary afferent activation and inactivation through AS is also reflected in the second-order neuronal activity, we applied AS onto the exposed dura mater, but AS neither changed the spontaneous ongoing activity nor the mechanically evoked activity of neurons. To control if TRPA1 activation is able to change the central neuronal activity, we repeated the experiments with the established TRPA1 agonist acrolein at two concentrations. There was again no significant change in ongoing activity, but the mechanically evoked activity decreased slightly after acrolein application. These results let us conclude that AS causes vigorous activation of primary meningeal afferents, followed by desensitisation, which is likely independent of TRPA1 activation. Second, the short-lasting activation of meningeal afferents by AS is obviously not sufficient to activate second-order neurons in the spinal trigeminal nucleus, whereas the TRPA1 agonist acrolein seems to transiently desensitise trigeminal afferents and hence spinal trigeminal neurons for mechanical stimuli applied to the meninges.

These results are partly consistent and partly inconsistent with previous results from related experiments. In one study of our group with a very similar in vivo setting, hydrogen sulphide (H_2_S), which reacts with endogenous nitric oxide (NO) to form nitroxyl (HNO), was topically applied onto the dura mater. H_2_S caused a short-lasting (two second) activation followed by a long-lasting decrease in activity in the majority of spinal trigeminal units [15]. The fact that we did not see a similar short-lasting activation following the application of AS in the present experiments may result from the extremely short-lasting action of HNO due to its high reaction velocity in the tissue [16]. Admittedly, the observed differences warrant careful interpretation, since we cannot exclude species differences, but the short-lasting effect of H_2_S on second-order neurons in rat is reminiscent of the short-lasting vigorous activation of primary meningeal afferents in mice followed by long-lasting inactivation seen in the present study. However, these responses are obviously largely independent of TRPA1, since they are not only maintained but even more pronounced in animals with functional TRPA1 deletion. Previously, our group has used calcium imaging to monitor the excitatory effect of Angeli’s salt on cultivated murine dorsal root ganglion neurons [11]. The calcium signal caused by AS was reduced in the presence of the TRPA1 receptor antagonist HC030031 and lost in neurons of TRPA1^−/−^ mice, showing its dependence on TRPA1 receptors. In another previous study of our group, again the rodent hemisected skull preparation was used to study the action of the TRPA1 agonist acrolein on primary meningeal afferents [17]. Acrolein caused CGRP release from the dura mater, which was significantly reduced by preapplication of HC03003. However, acrolein failed to cause action potentials but increased the electrical activation threshold of primary afferents, which indicates local activation without propagated activity. It may be concluded that TRPA1 agonists cause local activation in peripheral afferent terminals followed by rapid inactivation of TRPA1 receptor channels. The short-lasting activation is usually not strong enough to elicit action potential discharges, which could effectively activate the first synapse in the spinal system. On the other hand, the long-lasting inactivation silencing completely the primary afferents may be relevant as a brake for synaptic activity at the second order neurons and for holding down their excitatory level. This assumption is also consistent with previous results from our group, showing that permanent afferent input to the spinal trigeminal second-order neurons is necessary to hold up their ongoing activity [18].

Having shown that the vigorous activation of primary meningeal afferents by Angeli’s salt is obviously not or not alone dependent on TRPA1 receptor channels, we find that the question arises as to which other mechanisms come into account for this phenomenon. In the submucosa of the colon and the ileum, HNO formed by Angeli’s salt has been shown to activate basolateral K^+^ channels and the Na^+^-K^+^-ATPase [19], most likely via the sGC-cGMP (soluble guanylate cyclase—cyclic guanosine monophosphate) pathway [20], which results in smooth muscular relaxation [21]. Similar mechanisms have been described involving voltage-dependent K^+^ channels and ATP-sensitive K^+^ channels (K_ATP_) of smooth arterial muscle [22,23]. If a similar mechanism is implemented in primary meningeal afferents, this would result in hyperpolarisation, which could explain the long-lasting inactivation after Angeli’s salt application. Inactivating mechanism caused by K^+^ channel activation may also underlie the antinociceptive effect of AS in rodent models of inflammatory pain [24,25]. However, activation of any type of K^+^ channel cannot explain the short-lasting activation caused by Angeli’s salt, which may be speculated to be based on a quite different mechanism, possibly a direct action on voltage-dependent excitatory conduction channels.

Can we nevertheless have a hyperpolarising effect that is excitatory or that sensitises primary afferents? There is current discussion about the involvement of big-conductance calcium-activated potassium channels (BK_Ca_) and ATP-sensitive K^+^ channels (K_ATP_) in the initiation of migraine, which are both expressed in trigeminovascular tissues of different species, including humans [26,27]. Indeed, several mechanisms leading to an increased open probability of BK_Ca_ and K_ATP_ channels and direct openers of K_ATP_ channels have been demonstrated to cause not only pro-nociceptive responses and pain behaviour in animal models of migraine pain [28,29] but also delayed migraine-like attacks when infused into migraineurs [30,31]. The authors speculate that long-lasting hyperpolarisation through K_ATP_ activation would sensitise hyperpolarisation-activated cyclic-gated nucleotide channels (HCN) expressed from primary trigeminal afferents, although this issue is currently controversially discussed [32,33]. 

Taken together, nitroxyl delivered by Angeli’s salt has a dual effect on primary meningeal afferents in that a very short-lasting vigorous activation is followed by a long-lasting depression of activity and sensitivity, probably caused by different mechanisms. TRPA1 activation through nitroxyl seems to have a predominantly inactivating effect on the activity and sensitivity of meningeal afferents, which is reflected in an unchanged or decreased activity of second-order spinal trigeminal neurons.

## 4. Materials and Methods

All experiments were performed in accordance with the European Union’s regulations of care and treatment of laboratory animals (Council Directive 2010/63/EU). The protocols were reviewed by an ethics committee and approved by the District Government of Mittelfranken (Project Identification Code 54-2532.1-21/12). Animals were bred and held in the institute’s own animal house under a 12 h day–night cycle and had access to food and water ad libitum.

### 4.1. Hemisected Head Preparation and Experimental Setup

The following adult mice of both sexes (20–25 g body weight) were used for the in vitro experiments: C57BL/6 wild-type mice as well as TRPA1 knockout mice with functionally deleted TRPA1 receptors (TRPA1^−/−^) and TRPA1/TRPV1 double-knockout mice with functional deletion of both TRPA1 and TRPV1 receptors (TRPA1V1^−/−^) with a genetic background of C57BL/6. Breeding pairs of heterozygous TRPA1 knockout mice were obtained from Dr. Davis [34] and TRPV1 knockout mice from Dr. Corey [35], crossed and continuously backcrossed to C57BL/6. Homozygous animals were assessed by polymerase chain reaction using tail biopsy. Animals were euthanised by CO_2_ inhalation followed by decapitation. After removing skin, muscles, eyes, and the mandible, the skull was divided in the sagittal plane and the brain was removed without touching the parietal dura mater that lines the cranial cavity. The skull halves were kept in synthetic interstitial fluid, (SIF), consisting of 107.8 mM NaCl, 3.5 mM KCl, 0.69 mM MgSO_4_, 9.64 mM NaHCO_3_, 1.53 mM NaH_2_PO_4_, 7.6 mM Na-gluconate, 5.55 mM α-D(+)-glucose monohydrate, and 7.6 mM D(+)-sucrose, adjusted with carbogen (95% O_2_ and 5% CO_2_) to pH 7.4.

Each hemisected skull was transferred to a transparent acrylic chamber, embedded in Agar-Agar (Kobe I, Roth, Karlsruhe, Germany) with the cavity facing up, and covered with SIF (see Figure 1A). Under observation with a stereo-microscope (Leica MS5, Leica Microsystems, Wetzlar, Germany), the trigeminal ganglion was removed, and one of 2–3 meningeal branches of the mandibular nerve (V3), collectively called spinosus nerve, or the tentorius nerve, a branch of the ophthalmic nerve (V1), was freed from the surrounding dura and soaked with its free end into a glass pipette (tip diameter ≈20 µm) filled with SIF. The preparation was then continuously superfused with warm SIF in an open circulation with a flow velocity of 6 mL/min. The temperature was held constant by a flow-through Peltier element placed in series with the superfusion system and fed back by a thermocouple placed inside the tissue bath. The experiments were conducted at room temperature (24 °C), at which TRPA1 receptor channels are sensitive to stimuli and Angeli’s salt solution is sufficiently stable.

### 4.2. Recordings of Meningeal Afferent Activity and Chemical Stimulation

Action potentials of primary meningeal afferents were recorded via an AgCl wire inside the glass pipette (recording electrode) and an AgCl reference electrode immersed in the tissue bath. Receptive fields of meningeal afferents were localised by touching the cranial dura mater with von Frey filaments (10 mN, 0.2 mm diameter). Then, electrical pulses of 1 ms at a current of 15–50 µA were applied to the most sensitive spot of the receptive field, which produced action potentials at constant latency (see Figure 1A). The stimulator consisted of a glass capillary with a silver wire acting as an anode and a constantan wire (Ø 20 µm) serving as a cathode, both connected to a constant current stimulus isolator (A360 WPI, Sarasota, FL, USA). In some experiments, two or more discriminable action potentials with different latency and amplitude could be recorded. For each afferent fibre, the conduction velocity was assessed using latency and the distance between receptive field and recording site; the electrical threshold was defined as the 50% probability to evoke action potentials. Evoked action potentials and background activity were amplified with an Axopatch 200A (Molecular Devices (former Axon instruments), Sunnyvale, CA, USA), filtered (5 kHz low-pass), digitised (Micro 1401, Cambridge Electronic Design, Cambridge, United Kingdom), saved on a hard disk, and displayed on a computer monitor and a loudspeaker.

During the recording, electrical stimuli of 1.3–1.5 times the electrical thresholds were continuously applied at a frequency of 0.25 Hz to control the responsiveness of the afferent fibres (see Figure 1B). After a control period of 15 min, the flow-through was stopped two times for 5 min each to see if this procedure influences the afferent activity. Then, the circulation was again stopped and the vehicle NaOH (15 mM) was pipetted into the cranial cavity, followed by starting the flow-through again 5 min later to wash the vehicle out. The volume of the vehicle was the same as that of the following test substance, Angeli’s salt (AS) solution. After other 5 min, the same procedure was applied with AS solution, followed by wash-out and an observation period of 20 min. AS solution was freshly prepared immediately prior to the experiment by dissolving AS crystals with 15 mM NaOH to a stock solution of 1 mM. The volume of the applied AS stock solution was calculated to achieve an end concentration of 300 µM in the tissue bath.

### 4.3. Preparation for Recordings In Vivo and Experimental Setup

Adult male Wistar rats (310–410 g) were used for in vivo experiments. Animals were anaesthetised by inhaling oxygen-enriched air and 4% isoflurane (Forene, Abbott, Wiesbaden, Germany; evaporated by a Draeger Vapor 2000 system) in a closed box continued by application of 2% isoflurane through a tight mask. The eyes were protected by an ointment (Bepanthen, Bayer, Leverkusen, Germany). The body temperature was held at 37 °C with a feedback-controlled homeothermic system (Foehr Medical Instrument, Seeheim, Germany). The femoral artery and vein were exposed and catheterised for blood pressure measurements and intravenous injections. After quick intubation with a tube made from a thick venous catheter, the animals were ventilated with a mixture of oxygen-enriched room air (≈20% oxygen) and 2% isoflurane. Under these conditions, noxious stimuli (pinching of hindpaws and earlobes) failed to elicit motor reflexes or changes in systemic arterial pressure, which was between 90 and 100 mmHg (mean value). Expiratory CO_2_ levels were recorded and held at 4–5% by adjusting the ventilation volume and frequency, which was between 60 and 80 strokes/min. The head of the animal was fixed in a stereotaxic frame, and the skin overlying the skull and the neck region was opened. Using a dentist’s drill, we prepared a square-shaped cranial window under cooling with saline to expose the dura mater underlying the parietal bone on one side (see Figure 3A). During the whole experiment, the cranial window was covered with saline. The neck muscles were divided along the midline, the atlanto-occipital ligament and the underlying spinal dura mater were cut, and the medulla oblongata was exposed between the occipital bone and the atlas to gain access to the spinal trigeminal nucleus caudalis (SpVc). In order to terminate the experiments, a lethal dose of thiopentone (Trapanal, Nycomed, Konstanz, Germany) was intravenously injected.

### 4.4. Recordings of Spinal Trigeminal Activity and Chemical Stimulation

Custom-made carbon-fibre glass microelectrodes (impedance 1–2 MΩ) were inserted into the medulla on the side of the cranial window and moved at steps of 2.5 µm through the SpVc using a custom-made microstepper. The reference electrode was inserted subcutaneously nearby the recording electrode. Neurons with meningeal afferent input were identified by their responses to gentle mechanical probing of the exposed dura mater with von Frey filaments (2.9–11.8 mN), and meningeal receptive fields were characterised by their position and the mechanical threshold of the recorded neurons. Likewise, facial receptive fields were mapped by probing the temporal muscle; the skin of the ophthalmic (V1), maxillary (V2), and mandibular divisions of the trigeminal nerve; and the cornea. A custom-made mechano-stimulator with a blunt probe was adjusted to stimulate the dural receptive field with a constant force of about 1.5 times the threshold for 1 s every minute during the recording period (see Figure 3B). The recorded signals were band-passed and amplified (custom-made amplifier); then, they were digitised at a sampling rate of 20 kHz using a CED1402 controlled by Spike2 software (Cambridge Electronic design, Cambridge, United Kingdom). At the end of the experiment, a bipolar electrode was attached to the meningeal receptive field and the latency at just suprathreshold stimuli of 0.5 ms duration was assessed to calculate the conduction velocity of the primary afferent input. The position of the recorded neurons within the brainstem was approximately determined by the *x–y* position of the recording electrode and the recording depth.

When the observed neuronal activity of a unit was visibly stable, recording was started. After a control period of 15–20 min, the saline was carefully soaked from the cranial window, and 100 µL SIF was topically applied for 5 min, followed by washing for 1–2 min with SIF and application of 100 µL NaOH (vehicle for Angeli’s salt, 5 mM) for 5 min. After removal of the vehicle and washing with SIF, 100 µL of freshly prepared Angeli’s salt (300 µM in 5 mM NaOH) was applied for 5 min, then washed with SIF and the recording continued for 15–20 min. The same protocol was used for SIF (vehicle) and acrolein application (100 and 300 µM in SIF).

### 4.5. Data Processing and Statistics

The recorded data was analysed off-line using Spike2 version 6.08 (Cambridge Electronic Design, Cambridge, United Kingdom). Action potentials (APs) were identified using templates generated by stimulation of receptive fields. Templates of APs were defined according to the size and form of electrically evoked spikes and, in case of more than one clearly discriminable spike type (only in primary afferent recordings), assorted to different templates (see Figure 1B). Spikes were counted at 1 min intervals, while 5 min intervals were used for statistical testing.

#### 4.5.1. Primary Afferent Activity

The calculated activity of the primary afferent recordings includes the electrically evoked impulses (off-set of 15 impulses/min), which are not distinguishable from the other APs of the same template. Since electrically elicited spiking frequently failed (particularly after chemical stimulation, see the Results section), subtraction of this off-set from the whole activity would have resulted in negative spike counts, which would have impeded statistical calculation.

#### 4.5.2. Second Order Neuron Activity

The counted APs within 1 s prior to and 1 s after each stimulation minute were averaged and subtracted from the activity within the stimulation period to get the pure mechanically evoked activity during the 1 s stimulation period (APs/s). This evoked activity was subtracted from each 1 min activity to collect the pure spontaneous (ongoing) activity (APs/min).

#### 4.5.3. Statistics

The group size of experiments was determined on the basis of previous power calculations in similar experiments [15,17]. Animal or data points were not excluded from analysis. Data were processed using Microsoft Excel 2003 (Microsoft Corporation, Redmond, WA, USA) and Statistica 7.1 (StatSoft, Tulsa, OK, USA). Analysis of variance (ANOVA) with repeated measurements and Student’s *t*-test were used to analyse significance of effects. An error probability of *p* < 0.05 was assumed to indicate significance. Data are given as means ± standard error of the mean (SEM). Origin 7 (OriginLab Corp., Northampton, MA, USA) and Corel Draw ×7 (Corel Corp, Ottawa, ON, Canada) were used for graphical presentation.

## Figures and Tables

**Figure 1 ijms-23-02330-f001:**
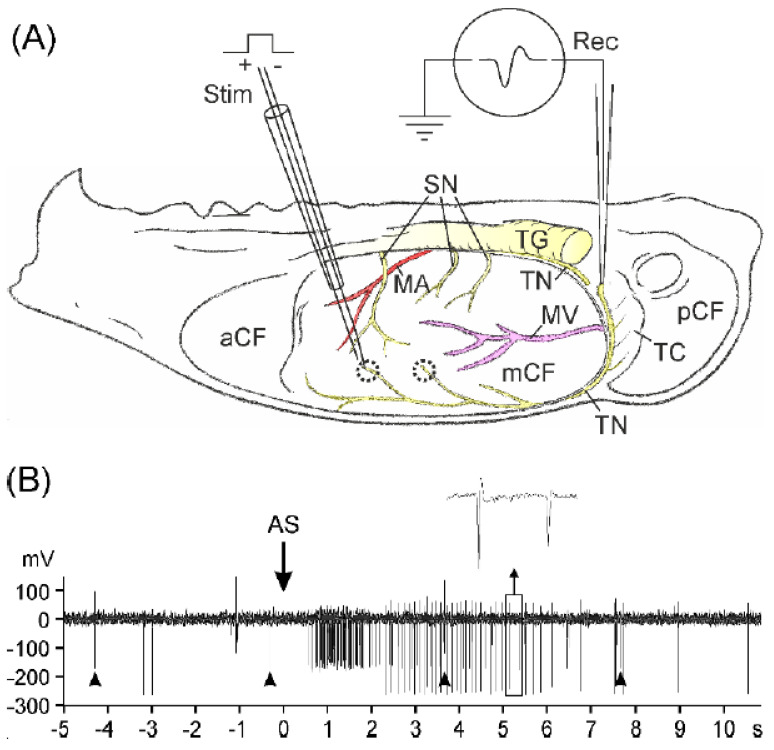
(**A**) Hemisected mouse skull preparation showing the approximate course of the spinosus nerves (SN) and the cut tentorius nerve (TN) with the recording electrode (Rec) attached. The receptive fields of two afferent fibres are dotted, the stimulation electrode delivers regular electrical pulses to one of them; aCF, mCF, pCF, anterior, middle, and posterior cranial fossa, respectively; MA, meningeal artery; MV, meningeal vein; TG, trigeminal ganglion; TC, tentorium cerebelli. (**B**) Part of a continuous recording from two afferent fibres (action potential shapes see inset) with their short-lasting responses to Angeli’s salt solution (AS), starting in one fibre about 0.5 s after the AS application, in the other one with a delay of about 2 s. Arrowheads show regularly electrically evoked action potentials of the fibre with the smaller amplitude.

**Figure 2 ijms-23-02330-f002:**
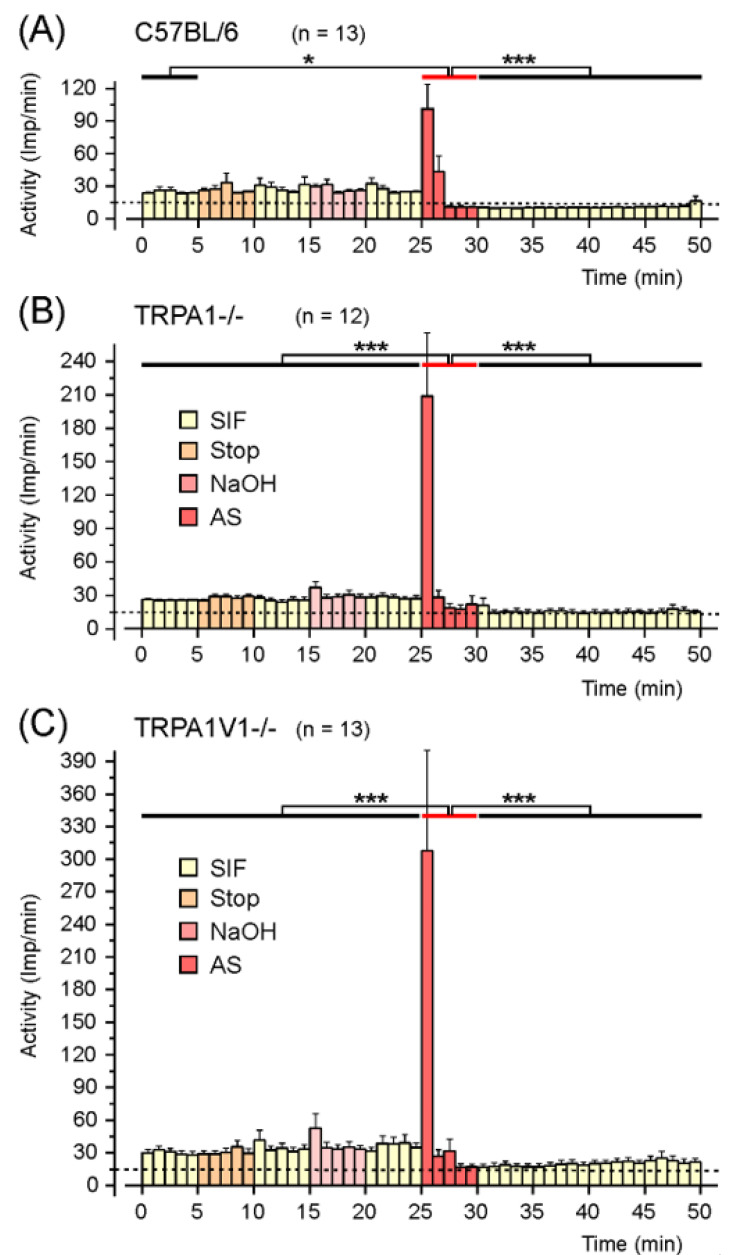
Mean activity (impulses/min) during the recordings from afferents of C57BL/6 wild-type mice, TRPA1 knockout mice, and TRPA1/TRPV1 double-knockout mice. Error bars represent standard error of the mean (SEM). The preparation was flushed with synthetic interstitial fluid (SIF) during the first 5 min control period, followed by a stop of the circulation; again, SIF supply, vehicle (NaOH), SIF, and Angeli’s salt solution (AS), followed by 5 min periods with SIF wash-out. In wild-type animals (**A**), the mean activity during AS application was significantly higher (*p* < 0.05, *) than the mean activity during the first control period (0–5 min); in the knockout animals (**B**,**C**), the increase in activity during AS application is highly significant (*p* < 0.0005, ***) compared with all periods prior to AS application. In all three mouse lines, the activity after AS application fell below baseline and was significantly lower than the activity during AS application (*p* < 0.0005, ***). The broken horizontal line indicates the off-set of 15 electrically evoked spikes/min added to the spontaneous action potentials; after the AS challenge, several fibres failed to be activated electrically, indicated by mean activity below this line.

**Figure 3 ijms-23-02330-f003:**
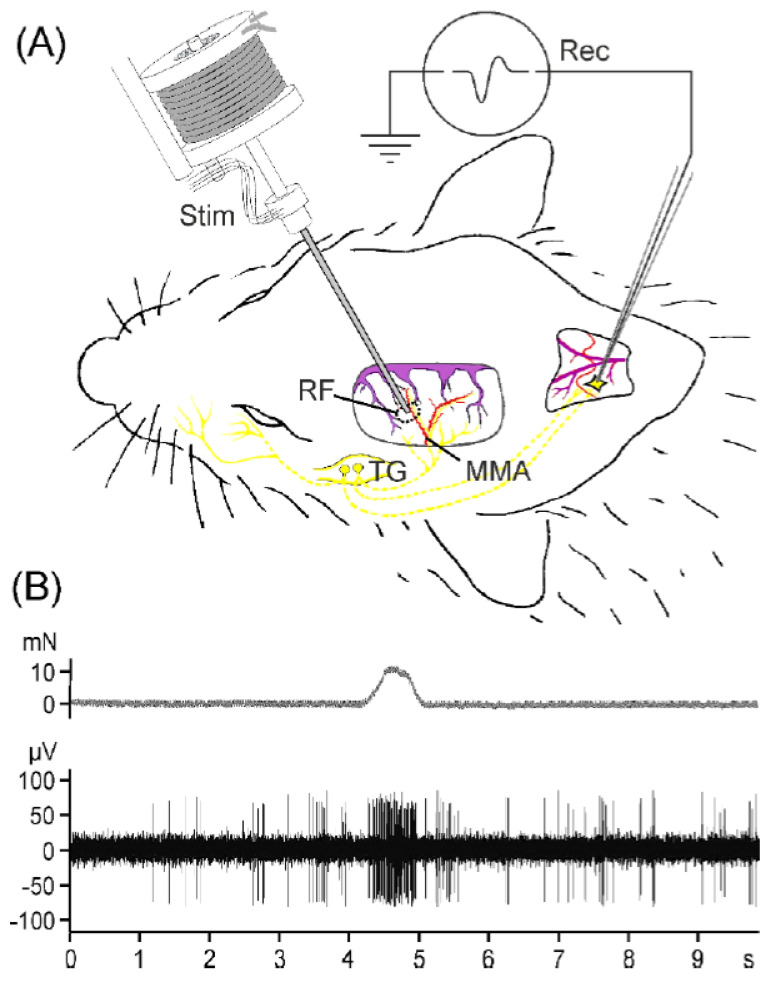
(**A**) Set-up for second-order neuron recording from the rat spinal trigeminal nucleus (Rec) and regular mechanical stimulation (Stim) of the receptive field (RF) in the exposed parietal dura mater. MMA, middle meningeal artery; SSS, superior sagittal sinus; TG, trigeminal ganglion. (**B**) Part of a continuous recording from a spinal trigeminal neuron (lower trace) responding to the 1 s mechanical stimulus applied at 1 min intervals (upper trace).

**Figure 4 ijms-23-02330-f004:**
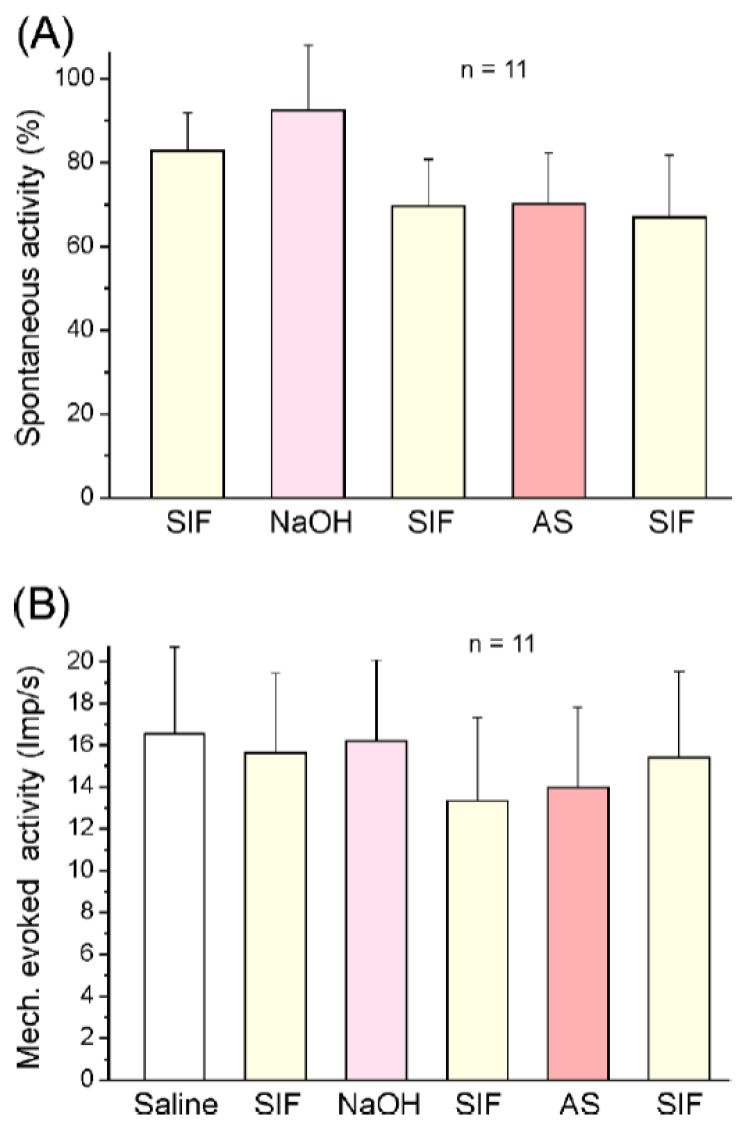
Mean spontaneous activity ((**A**), normalised) and mechanically evoked activity (**B**) at 5-min intervals of the sample of 11 neurons after application of SIF, NaOH (vehicle), and Angeli’s salt solution (AS). No significant differences; error bars indicate SEM.

**Figure 5 ijms-23-02330-f005:**
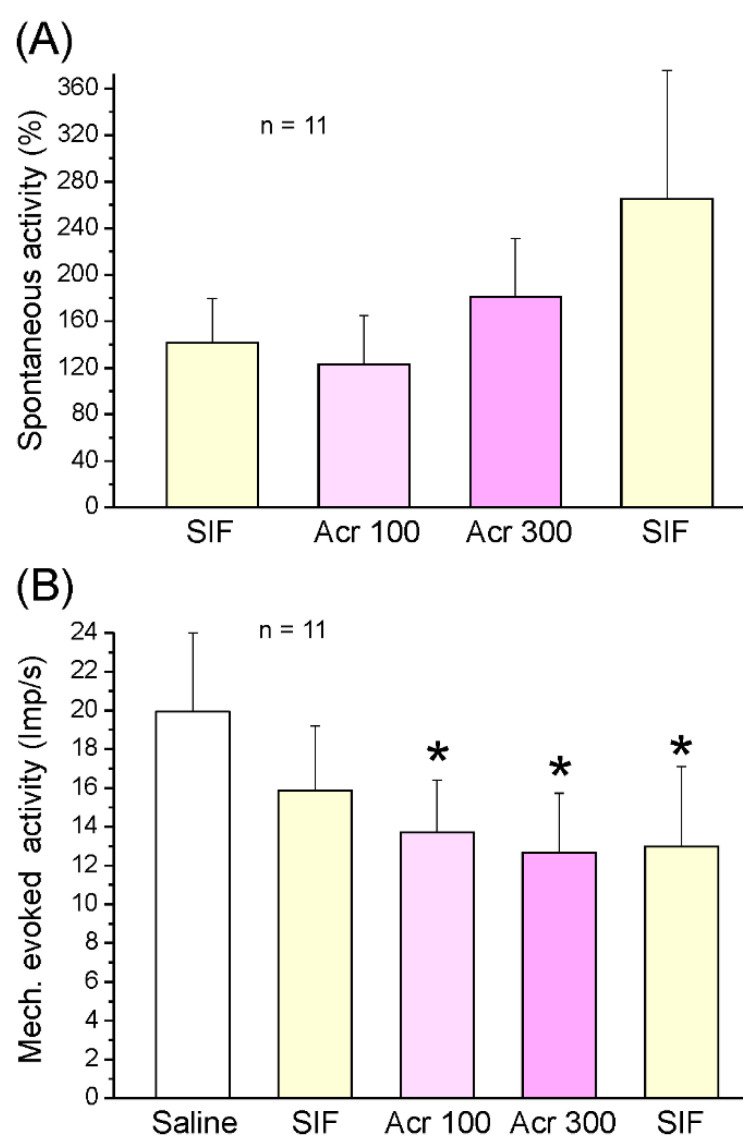
Mean spontaneous activity ((**A**), normalised) and mechanically evoked activity (**B**) at 5 min intervals of the sample of 11 neurons after application of SIF and acrolein (Acr) at 100 and 300 µM. * Significant difference to baseline (saline); error bars indicate SEM.

## Data Availability

Not applicable.

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
