# Peer review of "Nitroxyl Delivered by Angeli’s Salt Causes Short-Lasting Activation Followed by Long-Lasting Deactivation of Meningeal Afferents in Models of Headache Generation"

_ijms, 2022, doi:10.3390/ijms23042330_

Round 1

Reviewer 1 Report

The manuscript by StÓ§ckl et al. titled “The TRPA1 agonist nitroxyl delivered by Angeli’s salt causes short-lasting activation followed by long-lasting deactivation of meningeal afferents in models of headache generation” describes the effect nitroxyl donor (AS) exerts on meningeal afferents (MA) and spinal trigeminal neurons activity in rodent models of headache, and test whether it’s action require TRPA1 and TRPV1 channels. TRPA1 channels are located in afferents innervating the meninges and play a role in the nociception associated with headaches of different origin, including migraine pain. The effect of Angeli’s salt on spontaneous and evoked meningeal afferents activity was unexpected. First, as shown on preparations from KO animals, spontaneous increase in activity after Angeli’s salt application appeared to be independent of TRPA1/TRPV1 channels. Second, the evoked activity of meningeal afferents is depressed post Angeli’s salt application, and this effect is observed both in wt and KO animals, once again proving to be independent of TRPA1/TRPV1 expression. Finally, TRPA1 agonist acrolein, but not Angeli’s salt, mediates the depression of mechanically evoked activity of spinal trigeminal neurons. The observations are interesting and give valuable insight into TRPA1 channels and nitroxyl involvement in meningeal nociception.

The manuscript is coherent, methods and experiments description is precise, and discussion  addresses most observations made by Authors. There are some minor issues in the text/figures content that need to be clarified:

  1. Figure 2. What’s actually the subject of the comparison on the graphs? The graphs on A,B and C denotes the comparison (* or ***) between AS (red bars) and AS washout period? The description in the legend for Fig2A is saying: “The mean activity during AS application is significantly higher (p < 0.05, *) than the activity during the first control period” But what can be found on the graph is something else. The same for 2B and 2C. Please clarify. Was the difference between MA activity during first SIF and post AS washout application statistically significant which actually indicates desensitization?
  2. There are minor typos in the text (see pages 114, 121, 284, 295), therefore some editing or re-reading is suggested.
  3. The title is a kind of problematic. It recognizes Angeli’s salt as a “TRPA1 agonist” and then manuscript defines its effects on MA activity which are independent of TRPA1 channels. It may be beneficial for the paper if the title is reorganized and the phrase “TRPA1 agonist” omitted from the title.

Author Response

The manuscript by StÓ§ckl et al. titled “The TRPA1 agonist nitroxyl delivered by Angeli’s salt causes short-lasting activation followed by long-lasting deactivation of meningeal afferents in models of headache generation” describes the effect nitroxyl donor (AS) exerts on meningeal afferents (MA) and spinal trigeminal neurons activity in rodent models of headache, and test whether it’s action require TRPA1 and TRPV1 channels. TRPA1 channels are located in afferents innervating the meninges and play a role in the nociception associated with headaches of different origin, including migraine pain. The effect of Angeli’s salt on spontaneous and evoked meningeal afferents activity was unexpected. First, as shown on preparations from KO animals, spontaneous increase in activity after Angeli’s salt application appeared to be independent of TRPA1/TRPV1 channels. Second, the evoked activity of meningeal afferents is depressed post Angeli’s salt application, and this effect is observed both in wt and KO animals, once again proving to be independent of TRPA1/TRPV1 expression. Finally, TRPA1 agonist acrolein, but not Angeli’s salt, mediates the depression of mechanically evoked activity of spinal trigeminal neurons. The observations are interesting and give valuable insight into TRPA1 channels and nitroxyl involvement in meningeal nociception.

The manuscript is coherent, methods and experiments description is precise, and discussion  addresses most observations made by Authors. There are some minor issues in the text/figures content that need to be clarified:

1. Figure 2. What’s actually the subject of the comparison on the graphs? The graphs on A,B and C denotes the comparison (* or ***) between AS (red bars) and AS washout period? The description in the legend for Fig2A is saying: “The mean activity during AS application is significantly higher (p < 0.05, *) than the activity during the first control period” But what can be found on the graph is something else. The same for 2B and 2C. Please clarify. Was the difference between MA activity during first SIF and post AS washout application statistically significant which actually indicates desensitization?

Response: We are sorry for the unclear explanation of the legend. For clarification, we have added “(0-5 min)” to “the first control period” and “the increase in activity during the AS application is highly significant”. Also, we have added “In all three mouse lines, the activity after AS application fell below baseline and was significantly lower than the activity during AS application (p < 0.0005, ***)”. In addition, we have changed Figure 2 trying to indicate the significant differences more clearly.

2. There are minor typos in the text (see pages 114, 121, 284, 295), therefore some editing or re-reading is suggested.

Response: We tried to find the indicated typos together with some other ones and corrected them as far as we were aware of.

  1. The title is a kind of problematic. It recognizes Angeli’s salt as a “TRPA1 agonist” and then manuscript defines its effects on MA activity which are independent of TRPA1 channels. It may be beneficial for the paper if the title is reorganized and the phrase “TRPA1 agonist” omitted from the title.

Response: We have omitted “TRPA1 agonist” from the title.

We thank the reviewer for her/his competent review.

Reviewer 2 Report

This study is an interesting finding showing that activation of TRPA1 has an inhibitory function in the nociceptive trigeminal nervous system. I think this is an important finding in future migraine research. I have a few questions.

  • Consider whether the effects of acrolein are suppressed by TRPA1 antagonists and CGRP receptor antagonists.
  • Is it difficult to perform experiments using rats in experiments using wild type mice?

Author Response

This study is an interesting finding showing that activation of TRPA1 has an inhibitory function in the nociceptive trigeminal nervous system. I think this is an important finding in future migraine research. I have a few questions.

  • Consider whether the effects of acrolein are suppressed by TRPA1 antagonists and CGRP receptor antagonists.

Response: This a very good point. In our previous study by Denner et al. (ref. 17), the CGRP releasing effect of acrolein from the dura mater in the hemisected skull was significantly reduced by preapplication of the TRPA1 receptor antagonist HC030031. Likewise, in the same preparation used by Marics et al. (Cephalalgia 2017, DOI: 10.1177/0333102416654883) CGRP release upon acrolein was reduced by HC030031, and CGRP-dependent increase in meningeal blood flow evoked by application of acrolein was reduced by HC030031 and by the CGRP receptor antagonist CGRP(8-37). We have added to the discussion (2nd paragraph): “Acrolein caused CGRP release from the dura mater, which was significantly reduced by preapplication of HC03003...”

  • Is it difficult to perform experiments using rats in experiments using wild type mice?

Response: Ponting to the problem of using different species for the experiments is certainly justified. Unfortunately, in vivo recordings from second order neurons innervating the meninges in mice cannot yet sufficiently performed. We have addressed the problem in the Discussion (2nd paragraph) adding: “Admittedly, the observed differences warrant careful interpretation, since we cannot exclude species differences …”

We thank the reviewer for the important points indicated to improve our manuscript.

Round 2

Reviewer 2 Report

Thank you for adding / correcting the submitted paper.